# Quality Evaluation of *Polygonatum cyrtonema* Hua Based on UPLC-Q-Exactive Orbitrap MS and Electronic Sensory Techniques with Different Numbers of Steaming Cycles

**DOI:** 10.3390/foods13101586

**Published:** 2024-05-20

**Authors:** Mengjin Wang, Jiayi Hu, Xiaoya Hai, Tianzhuo Cao, An Zhou, Rongchun Han, Lihua Xing, Nianjun Yu

**Affiliations:** 1Department of Biopharmaceuticals, College of Pharmacy, Anhui University of Chinese Medicine, Hefei 230012, China; wmj12318@163.com (M.W.);; 2Anhui Province Key Laboratory of R&D of Chinese Medicine, Anhui Province Key Laboratory of Chinese Medicinal Formula, Hefei 230012, China; 3MOE-Anhui Joint Collaborative Innovation Center for Quality Improvement of Anhui Genuine Chinese Medicinal Materials, Hefei 230012, China

**Keywords:** *Polgonatum cyrtonema* Hua, electronic sensory, UPLC-Q-Exactive Orbitrap MS, quality evaluation

## Abstract

In this study, electronic sensory techniques were employed to comprehensively evaluate the organoleptic quality, chemical composition and content change rules for *Polygonatum cyrtonema* Hua (PCH) during the steaming process. The results were subjected to hierarchical cluster analysis (HCA), principal component analysis (PCA) and orthogonal partial least squares discriminant analysis (OPLS-DA). These analyses revealed, from a sensory product perspective, overall differences in colour, odour and taste among the samples of PCH with different numbers of steaming cycles. Using the UPLC-Q-Exactive Orbitrap MS technique, 64 chemical components, including polysaccharides, organic acids, saponins and amino acids were detected in PCH before and after steaming. The sensory traits were then correlated with the chemical composition. From the perspectives of sensory traits, chemical composition, and multi-component index content, it was preliminarily deduced that carrying out five cycles of steaming and sun-drying was optimal, providing evidence for the quality evaluation of PCH during the steaming process.

## 1. Introduction

*Polygonatum cyrtonema* Hua (PCH), belonging to the family Asparagales, is a medicinal and edible perennial plant. Plants of the same genus, including *P. kingianum* and *P. cyrtonema*, have also been used as substitutes for *P*. *Sibiricum* [1]. The functions of PCH include nourishing the liver and kidneys and promoting longevity [2,3]. In recent years, many scholars have discovered that PCH contains polysaccharides, steroidal saponins, flavonoids, alkaloids, and other ingredients [4,5,6,7]. PCH needs to be concocted before it can be used in medicine. Its main purposes are to reduce toxicity, increase effectiveness, eliminate throat irritation, enhance the function of tonifying the spleen, moistening the lungs, benefiting the kidneys, and enhancing biological activity [6,7].

There are numerous records on the methods for processing PCH, including single steaming, wine steaming, and nine cycles of steaming and sun-drying [8]. The “Nine Steaming Nine Sun-drying” method has appeared most frequently in successive herbal medical books, and has been widely used in the history of PCH processing methods. In ancient times, the determination of the endpoint of the “Nine Steaming Nine Sun-drying” process for PCH was performed using the naked eye to observe the changes in characteristics during the steaming process. The decision on how many cycles of steaming to perform was subjective, leading to speculation that in ancient times, “nine” simply indicated repeated steaming, rather than a specific requirement to steam nine times [9].

Current research on steamed *Polygonati rhizoma* has focused on its chemical composition, pharmacological activity, single-trait indexes and compositional correlations. Lanqi Zhou et al. [10] demonstrated substantial improvements in the anti-oxidative, anti-inflammatory and anti-hyperglycaemic effects of *Polygonatum kingianum* Coll and Hemsl (PK) water extracts processed with nine cycles of steaming and sun-drying compared with crude PK in cell models. Lian-lin Su et al. [11] showed that PCH processed with six cycles of steaming and sun-drying and PCH processed with nine cycles of steaming and sun-drying could significantly enhance the immune activity of the organism, improve the imbalance of intestinal flora in immunosuppressed mice, and increase the content of intestinal short chain fatty acids. It is noteworthy that processing PCH with six cycles of steaming and sun-drying has a better effect on the improvement of the immune activity of the organism. Zheng Xiaoqian et al. [12] investigated the correlation between colour changes and various chemical constituents during the process of nine cycles of steaming and sun-drying of PCH, and found that five cycles of steaming and sun-drying reached the standard of steaming. In general, most current studies are focused on a certain point and lack comprehensive analysis. As a result, there is no clear conclusion on the quality evaluation of PCH with different numbers of steaming cycles.

Intelligent electronic sensor technology can mimic the human senses of sight, smell and taste. The data indicators can be output through the computer to avoid human error factors like subjectivity and fatigue. The sensory quality of herbs is closely related to their appearance, colour, odour, and texture. The characteristics and intensity of colour, smell, and taste reflect the quality of traditional Chinese medicine, to a certain extent [13,14].

The primary methods for analysing the chemical composition of traditional Chinese medicine include ultra-performance liquid chromatography (UPLC), nuclear magnetic resonance (NMR), and mass spectrometry (MS). In recent years, with the advancement of high-resolution mass spectrometry, sensitivity and resolution have greatly improved. LC-MS has the advantages of high sensitivity, high stability, and high throughput, and is increasingly utilised in the analysis of complex systems in traditional Chinese medicine [15,16]. Wang Y et al. [17] effectively identified 54 compounds, including flavonoids, amino acids, organic acids, terpenoids, coumarins, lignans, and other compounds in Chinese herbal tablets over different fermentation periods, and determined the content of their characteristic components. Wang H et al. [18] identified 31 compounds in Cullen Corylifolium using LC-MS/MS, of which eight were representative, and 62 compounds using GC-MS, including hydrocarbons, terpenoids and coumarins. These studies provide a crucial foundation for further research on their pharmaceutical activity and quality control. This study utilised the UHPLC-Q-Exactive-MS technique for the rapid resolution of chemical constituents to study the differences in chemical composition profiles in PCH samples subjected to different numbers of steaming cycles, with the aim of evaluating the qualities of PCH with different numbers of steaming cycles.

## 2. Materials and Methods

### 2.1. PCH Samples

All PCH herbs were supplied by Jiuhua Chinese Herbal Medicine Science and Technology Co. (Chizhou, China), and the employed steaming equipment was supplied by Baifeng Trading Co. (Linyi, China) PCH raw herbs were harvested in the fall and winter seasons, and then washed, boiled in water, removed from the water, and dried in the sun. These samples were marked as S0. The samples were placed into the steaming equipment. Steaming began at a high heat, so that the water vapor filled the entire steam unit, first using 100 °C for 1 h, then 60 °C at a constant temperature for 50 min, and finally removing for drying. This cycle was repeated eleven times, referred to as “eleven steaming systems”. Samples were taken after each steaming process and marked as S1~S11.

### 2.2. Sensory Technique Samples and Methods

#### 2.2.1. Appearance Traits and Sensory Evaluation

The colour, texture, odour, and taste profiles of the products were rated separately from both medicinal and culinary perspectives, in conjunction with the sensory evaluation of the food. The assessment panel consisted of 10 members who had been professionally trained. Referring to the criteria in Table 1, each steamed sample was placed on a white background. Its colour, lustre, and texture were observed under natural light, and its smell was evaluated to determine if there were any off odours. Afterwards, the taste was evaluated, and the PCH was scored for appearance properties and sensory evaluation with a total score of 100. The sensory properties were then documented and described.

#### 2.2.2. Electronic Eye Detection

The dried herb samples were powdered, and then passed through a No. 4 sieve. After stabilising the power, the lens exposure and focal length were corrected and calibrated with a chromaticity plate placed in the tray, using a 5 mm lens with top and bottom illumination. A specific amount of sieved powder from each PCH sample was spread on a slide. The slide was then placed on the white-bottom plate of the instrument to capture images. The position was adjusted, and pictures were taken in parallel three times. The chromaticity space parameters of the samples during the steaming process of PCH are represented by *L** (*L** = Luminance, 100 means white, 0 means black), *a** (+*a** means red, −*a** means green), *b** (+*b** means yellow, −*b** means blue). The Δ*E** value is calculated according to the following equation:$${\Delta E}^{*}=\sqrt{{\left(L_{0}^{*}-L^{*}\right)}^{2}+{\left(a_{0}^{*}-a^{*}\right)}^{2}+{\left(b_{0}^{*}-b^{*}\right)}^{2}}$$

#### 2.2.3. Electronic Nose Detection

The PEN3 electronic nose sensor response reflects the adsorbed gas state of chemical molecules and subsequent surface chemical reactions or physical effects resulting from changes in resistivity and conductivity [19]. The response value is calculated as G/G0. If G/G0 ≤ 1, it means that the sample has no response to this sensor. The characteristics are detailed in Table 2. Each sample was taken, placed in a 50 mL centrifuge tube, sealed with a film, and left for 30 min to enrich before testing. The electronic nose sampling time was 1 s per group, the sensor self-cleaning time was 80 s, the sensor zeroing time was 5 s, the injection flow rate was 400 mL/min, and the analysis sampling time was 100 s. Three parallels were made for each sample, and the stable signal obtained at 80 s from the sensor was taken for analysis.

#### 2.2.4. Electronic Tongue Detection

The electronic tongue detected eight attributes of PCH samples: sourness, bitterness, astringency, freshness, saltiness, and sweetness. Six sensors, C00, AE1, CA0, CT0, AAE, and GL1, corresponded to bitterness, astringency, sourness, saltiness, freshness, and sweetness, respectively, in addition to bitter aftertaste, astringent aftertaste, and richness. The types of electronic tongue sensor responses are shown in Table 3.

A measure of 1.5 g of sample powder was weighed in a centrifuge tube, 50 mL of distilled water was added to soak for 30 min, and the sample was placed into a sonicator and sonicated for 1 h. After sonication, the sample was centrifuged for 10 min at a speed of 4000 r/min, and the supernatant was collected. A measure of 50 mL of distilled water was added, the extraction was repeated once following the above conditions, the supernatant was centrifuged, and the obtained supernatant was combined twice. The sensor was first cleaned in the cleaning solution for 90 s. The reference solution was then cleaned for 120 s, followed by cleaning with another reference solution for 120 s. The sensor was zeroed at the equilibrium position for 30 s. The test was cycled four times, the first cycle was removed, and the average of data from the last three times were taken as the test results.

All experiments using the electronic eye, electronic nose and electronic tongue needed to be measured three times in parallel and averaged. Origin 2021, SIMCA 14.1 and IBM SPSS 26.0 were used for image plotting and data processing.

### 2.3. UHPLC-Q-Exactive-MS Sample Preparation and Methods

A Vanquish (Thermo Fisher Scientific, Waltham, MA, USA) UPLC system was used with an ACQUITY UPLC BEH C18 (2.1 mm × 100 mm, 1.7 µm) column from Waters. The injection volume was 5 μL and the flow rate was 0.5 mL/min. The mobile phase consisted of 0.1% formic acid aqueous solution (A) and 0.1% formic acid acetonitrile solution (B). The elution gradient was programmed as follows: 0~11 min, 85~25% A; 11~12 min, 25~2% A; 12~14 min, 2~2% A; 14~14.1 min, 2~85% A; 14.1~16 min, 85%~85% A.

The MS system was based on a Thermo Fisher Scientific Orbitrap Exploris 120 mass spectrometer (workstation Xcalibur 4.3), and the Full MS-dd-MS2 scan mode (*m*/*z* 100–1500 Da) was used, with data acquired under positive and negative ion conditions, resulting in IDA-based acquisition modes of MS as well as MS/MS data. The sheath gas flow rate was 35 Arb, the auxiliary gas flow rate was 15 Arb, the ion transfer tube temperature was 350 °C, the vaporiser temperature was 350 °C, the full MS resolution was 60,000, the MS/MS resolution was 15,000, the collision energy was 16/32/48 (NCE mode), and the spray voltage was either 4 kV (+) or 3.8 kV (−).

Based on the Chinese and English literature, and querying Chemspider, Pubchem, TCMSP, HMDB, KEGG, MassBank and other online databases, we searched for the reported chemical constituents in PCH and established a database of chemical constituents of PCH, including their compound names, molecular formulae, molecular weights, and chemical structures. The molecular formulae of the chromatographic peaks were deduced using the Thermo Xcalibur 2.1 workstation, and the raw data were imported using XCMS software for retention time correction, peak identification, peak extraction, peak integration, and peak alignment, with the mass spectral deviation range of δ ≤ 5 ppm. The preprocessed mass spectral data information was compared with the secondary mass spectral fragmentation ions of the database and the chemical formula of the compound, and the chemical formulae of the compound were identified. We also searched for the possible chemical formulae and cleavage fragments of the compounds in combination with the literature and repeatedly compared and verified the identified compositions.

### 2.4. Data Statistics and Analysis

Data are expressed as the mean ± standard deviation (SD) of 3 independent experiments; statistical analysis was performed with SPSS Statistics 26 software, using least significant difference (LSD) multiple comparisons, principal component analysis (PCA), and orthogonal partial least squares discriminant analysis (OPLS-DA). Significant difference between different groups was considered at *p* < 0.05. Correlation analysis was performed using the Pearson correlation coefficient. Plots were performed using Origin 2022, Simca 14.1.

## 3. Results and Discussion

### 3.1. Sensory Quality Analysis

#### 3.1.1. Appearance Properties and Organoleptic Evaluation of PCH with Different Numbers of Steaming Cycles

The appearance of PCH after industrial steaming is documented in Figure 1 which clearly illustrates the yellow-white colour of the multifloral yellow essence before steaming (S0). The colour changed from brownish brown to black as a result of between one and five cycles of steaming (S1–S5). After five cycles of steaming and sun-drying, the PCH samples all appeared uniformly black, with the naked eye gradually unable to distinguish the difference; however, the subsequent colour differences also needed to be detected using the electronic eye for the colour test. The sensory scores and evaluation of descriptive results are shown in Table 4. From Table 4, it can be observed that as the number of steaming cycles increased, the sensory scores of PCH showed a pattern of initially increasing and then decreasing. The scores of S0 and S1 of PCH were below 60, indicating poor sensory evaluation. With the change in the number of steaming cycles, the scores gradually increased, and the flavour improved over time. Notably, the scores of PCH S5–S8 were higher, and the difference was not significant. However, the scores of No. S9–S11 PCH gradually decreased with the increasing number of steaming cycles, and the organoleptic evaluation results deteriorated over time. As can be seen from Table 4, the texture of PCH in the steaming process changed from hard to soft and sticky; raw products did not have a distinct smell, but as the number of steaming cycles increased, the smell gradually became stronger, and then a burnt aroma or even burnt flavour was exuded. Simultaneously, the overall taste was changed, and the initial bitter and astringent numbness of the tongue was gradually eliminated, with the sweetness enhanced.

#### 3.1.2. Variation in Colour Parameters of PCH with Different Numbers of Steaming Cycles

In the first stage, the colour of PCH was found to change gradually from yellowish white to blackish black with different numbers of steaming cycles, but in the later stage, the change in colour was not obvious, and the recognition of the colour change by the naked eye was reduced. As can be seen from Figure 2a, a* and b* are both positive values, representing that the external colour of PCH is generally between yellow and red. As the number of steaming cycles increased, the a* value showed a gradual increase, indicating that the colour of the sample gradually changed to red, while the b* value gradually decreased, indicating that the orange-yellow colour gradually faded away. The S0 sample had the highest L*, which gradually decreased with the number of steaming cycles, indicating a gradual decrease in brightness, which manifested as a deepening of the brown colour. The larger the ΔE* value, the larger the colour difference; conversely, the smaller the value, the smaller the measured colour difference. Taking the raw PCH S0 as a reference, S1 had the lowest value of ∆E*. The values of ∆E* of the samples all increased significantly with the number of steaming cycles, indicating that the samples steamed once were closest to the colour of the raw product. In addition, the fluctuation of ∆E* values of S5–S11 of PHC was small, indicating that the change in its blackish colour was gradually stabilised.

#### 3.1.3. Changes in Odour of PCH with Different Numbers of Steaming Cycles

As can be seen from Figure 2b, the response values of the 10 sensors show significant differences with the increase in the amount of evaporation, and the differences are mainly reflected in the four sensors W1W, W2W, W5S, and W1S. No response or a low response was found on W1C, W3C, W6S, W5C, W2S, and W3S, which can be ignored. The response values of W1W, W2W, W5S, and W1S were higher, and the response values were all greater than 1, which represented the main components of the odour of PCH. It is presumed that the differences in the odour of PCH during industrial steaming are mainly a result of the differences in volatiles such as terpenes, inorganic sulphides, aromatics, nitrogen oxides, alkanes, etc. The response values of the four sensors of S0 were all low, less than 2, indicating that raw PCH contained less odour information. The response values of W1W, W2W and W5S increased with the number of steaming cycles, while W1S exhibited a decreasing trend, until the response values of S11 PCH were significantly higher than those of the samples on the W1W, W2W, W5S and W1S sensors. It is assumed that with the increase in the number of steaming cycles, the high temperature leads to the continuous decomposition of proteins, producing sulphides, methane and nitrogen oxides. Therefore, the burnt aroma of the PCH after steaming may be attributed to these components.

#### 3.1.4. Changes in Flavour of PCH with Different Numbers of Steaming Cycles

The electronic tongue test simulates the state of the human mouth when only saliva is present and uses a reference solution to determine taste values. Taste-free points exist for sour and salty flavours, with −6 for salty, −13 for sour, and 0 for the rest of the tastes. When a specific taste response value is less than the no-taste point, it indicates the taste does not exist, and vice versa, the corresponding taste is present. As can be seen from Figure 2c, the order of response values in S0 was as follows: sweetness > freshness > bitterness > astringency. With the increase in the number of steaming cycles, the four flavour response values showed a gradual declining trend. Six cycles of steaming did not appear to significantly reduce this change. The response values for bitter aftertaste, astringent aftertaste and richness were all lower in the samples. There was no significant change in the response values with an increase in the number of steaming cycles. The sour flavour did not show a response in S0–S5, but did show a response value in S6. Furthermore, the response value increased as the number of steaming cycles increased. The aforementioned changes suggest that the unpleasant taste decreased after repeated steaming, while the carbonyl-ammonia condensation reaction occurred between aldoses and ketoses, which are sweet sugars, and amino acids and proteins, which have fresh flavours [20].

#### 3.1.5. Statistical Analyses

Hierarchical cluster analysis (HCA) [21] was conducted using the information from the electronic eye, nose, and tongue indicators of PCH samples with different numbers of steaming cycles. The cluster analyses among the samples are shown in Figure 3. The results showed that when the distance was 10, the samples were divided into two main classes. S0 was clustered into one major class alone, while S1, S2, S3, and S4 were clustered into another major class. Additionally, S5, S6, S7, S8, S9, S10, and S11 were clustered into a separate major class. Further analysis shows that when the distance is 5, the samples are divided into five classes in total: S0; S1; S2, S3, and S4; S5, S6, S7, and S8; and S9, S10, and S11. The results indicate that HCA can determine whether PCH has been steamed or not based on the organoleptic information indexes, and it can group together samples with similar organoleptic information during the steaming process of PCH. However, HCA cannot provide more detailed information regarding the differences between the samples of PCH and the organoleptic indexes. This necessitates multivariate statistical analysis, which combines principal component analysis (PCA) and orthogonal partial least squares discriminant analysis (OPLS-DA).

The dataset composed of three intelligent sensory technology indicators was fitted with multi-source information and analysed; it can be seen from Figure 4a that the $R_{X}^{2}$ is 0.996, representing the rate of explanation of the dimensionality-reduced data compared to the original data. The closer the value is to 1, the more desirable it is, and a value greater than 0.5 indicates that the model is more effective. The sum of PC1 and PC2 reached 84.3%, representing 84.3% of the information from all indicators. The greater the distance between samples, the greater the overall sensory difference. Principal component analysis revealed differences in colour, odour and taste among the steamed products of PCH. S0 is outside the confidence ellipse and is an outlier, indicating that the organoleptic information indexes of the raw sample differ greatly from those of the rest of the samples, while S9, S10, and S11 are clustered, indicating that the organoleptic information of the three samples is, to a certain extent, similar. In addition, this study also compared the models of single techniques for sensory quality evaluation of PCH; the results are shown in Figure 4b–d. The $R_{X}^{2}$ in the PCA of e-eye samples was 0.985; in the PCA of e-nose samples, it was 0.995; in the PCA of e-tongue samples, it was 0.917.

The outliers (S0) were observed in the PCA, and the OPLS-DA was performed after removing the outliers. As shown in Figure 5a, after excluding S0, the cumulative statistics of the OPLS-DA model were as follows: $R_{X}^{2}$ = 0.999, $R_{Y}^{2}$ = 0.942, and Q^2^ = 0.83. All values exceeded 0.5, suggesting an improved predictive capability of the model for the sensory information indices of the PCH steaming process. To further determine the stability of the model, the replacement test (*n* = 200) was used to validate the model. As shown in Figure 5b, all the green R2 values and blue Q2 values on the left side are lower than the original points on the right side. The blue regression line for the Q2 points intersects the vertical axis, indicating the validity of the original model. Based on the variable importance projection (VIP) values used to assess the variability of sensory information during the steaming process of PCH, it can be observed from Figure 5c that 11 indicators, namely, richness, W1S, a*, W3S, W6S, bitterness, astringency, bitter aftertaste, W1W, W5S, and W5C, were identified as having the most significant impact on the organoleptic evaluations of samples before and after PCH steaming, based on the criterion that the VIP value was >1.

### 3.2. UHPLC-Q-Exactive-MS Chemical Composition Analysis

#### 3.2.1. Identification of Chemical Constituents of PCH with Different Numbers of Steaming Cycles

The aqueous extract of PCH was analysed using the UPLC-Q-Exactive Orbitrap MS technique with different numbers of steaming cycles, and TIC plots of PCH aqueous extract were obtained in positive and negative ion modes (ion flow diagram in Appendix A), which are shown in Figure 6. A total of 64 chemical components were identified by preprocessing the mass spectrometry data information, matching it with the database and combining it with fragment ion information from the literature. Among them, there were nine organic acids, eight flavonoids, seven alkaloids, eight amino acids, seven saponins, and twenty-five other compounds. The cleavage information of the identified compounds is shown in Table 5.

#### 3.2.2. Changes in Chemical Composition of PCH during Steaming Process

Figure 7 shows that there is no obvious difference in the types of chemical constituents before and after PCH steaming, but the relative content of each constituent varies greatly in the middle (detailed identification and analysis of compositions in Appendix A). The total amount of saponins gradually decreased with the increase in the number of steaming cycles, possibly due to high-temperature cleavage during the steaming process and the formation of secondary glycosides or glycosides. The relative content of amino acids and organic acids also showed a decreasing trend, possibly attributed to their solubility in water. The structure of these compounds is easily damaged, decomposed, or denatured at high temperatures. Flavonoids and alkaloids exhibited a pattern of initial increase followed by a decline, likely attributed to the elevated temperature during the steaming process, facilitating dissolution. However, with repeated steaming, glycosides may undergo high-energy bombardment, leading to glycosyl fracture. The relative contents of other small molecules, such as fatty acids and phenylpropanoids, showed a significant increasing trend.

## 4. Association Analysis

### 4.1. Correlation between Odour and HPLC Analysis

A positive correlation was observed between W1W, W2W, and W5S and Tyrosine, 5-hydroxymethylfurfural, Glycine, Phenylalanine, 2-Aminophenol, and Phenylacetic acid (r > 0.6), while a negative correlation was observed between W1W, W2W, and W5S and Proline, Valine, and Arginine (r < −0.7) (Figure 8). W1S was positively correlated with Proline and Valine, and negatively correlated with Tyrosine, 5-hydroxymethylfurfural, Glycine, D-ornithine, and 2-Aminopheno.

### 4.2. Correlation between Flavour and HPLC Analysis

A positive correlation was found to exist between umami, bitterness, and astringency and Proline, Valine, and Arginine (r > 0.7), while a negative correlation was observed between umami, bitterness, and astringency and Tyrosinine, Glycine, D-ornithine, 5-hydroxymethylfurfural, Vanillin, and 2-Hydroxy-4-methylbenzaldehyde (r < −0.7) (Figure 9). Sourness was positively correlated with Tyrosinine, Glycine, D-ornithine, 5-hydroxymethylfurfural, Vanillin, and 2-Hydroxy-4-methylbenzaldehyde (r > 0.7) and negatively correlated with Proline, Valine, and Arginine (r < −0.7).

## 5. Conclusions

The results of this study showed that, with an increase in the number of steaming cycles, PCH gradually changed from yellow-white to brown. The odour response value increased after steaming, and the scent gradually became more intense, displaying sweet and smoky notes. Sweetness, freshness, bitterness and astringency gradually decreased, and sourness emerged after the sixth steaming. Multivariate statistical analysis, after applying the three techniques, indicated an overall difference in colour, odour and taste among the samples. The assessment panel results returned higher scores after the fifth cycle of steaming. Combining these results with those of the electronic senses from a sensory quality evaluation standpoint, the Five Steaming Five Sun-drying method successfully achieved the steaming objective [80].

The chemical composition profile of the extract of PCH from the steaming process was analysed using the UHPLC-Q-Exactive-MS technique. A total of 64 chemical constituents were identified, including eight flavonoids, nine organic acids, seven alkaloids, eight amino acids, seven saponins and twenty-five other compounds. The chemical compositions of PCH samples with different numbers of steaming cycles were similar, all dominated by flavonoids, organic acids, alkaloids, saponins, etc. However, the relative contents of each type of component differed considerably.

To sum up, the processing times of the traditional “Nine Steaming Nine Sun-drying” method are longer, affecting the quality of industrial products and production costs. Reducing the number of steaming cycles in the PCH process can effectively save on economic and time costs, ultimately enhancing industry efficiency [81]. Therefore, based on sensory quality traits, chemical composition, and multi-component index content, the preliminary inference suggests that five cycles of steaming and sun-drying is optimal. This method reduces the throat irritation cause by the raw product, making it more conducive to the future development of PCH and industrial processing.

## Figures and Tables

**Figure 1 foods-13-01586-f001:**
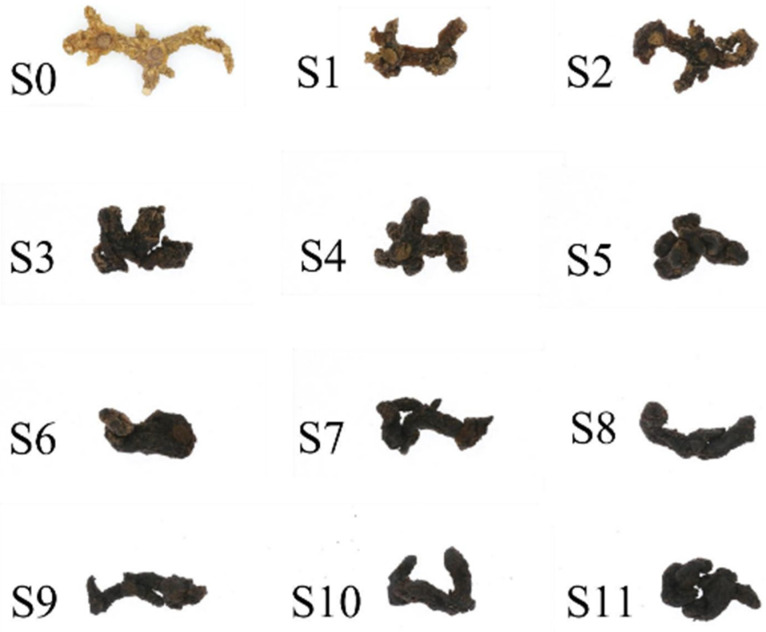
Industrialisation of steaming samples of PCH.

**Figure 2 foods-13-01586-f002:**
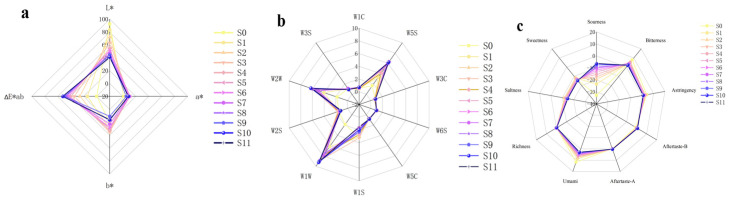
Sensory evaluation results of PCH industrialised steaming. (**a**) Electronic eye radar analysis of PCH industrialised steaming; (**b**) electronic nose radar analysis of PCH industrialised steaming; (**c**) electronic tongue radar analysis of PCH industrialised steaming.

**Figure 3 foods-13-01586-f003:**
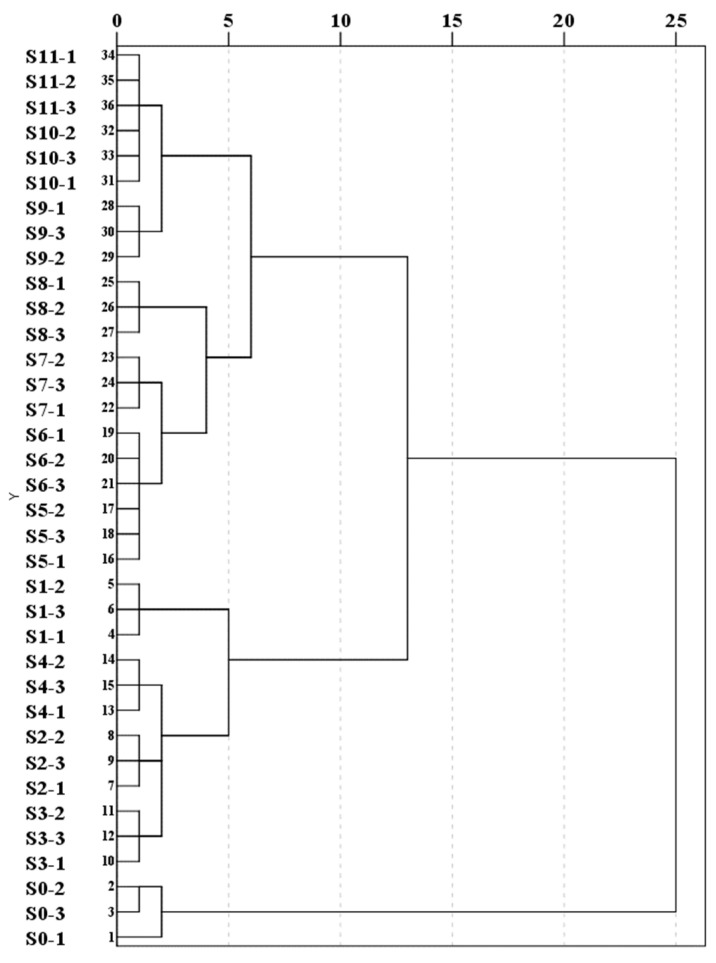
Cluster analysis chart of PCH after industrialised steaming.

**Figure 4 foods-13-01586-f004:**
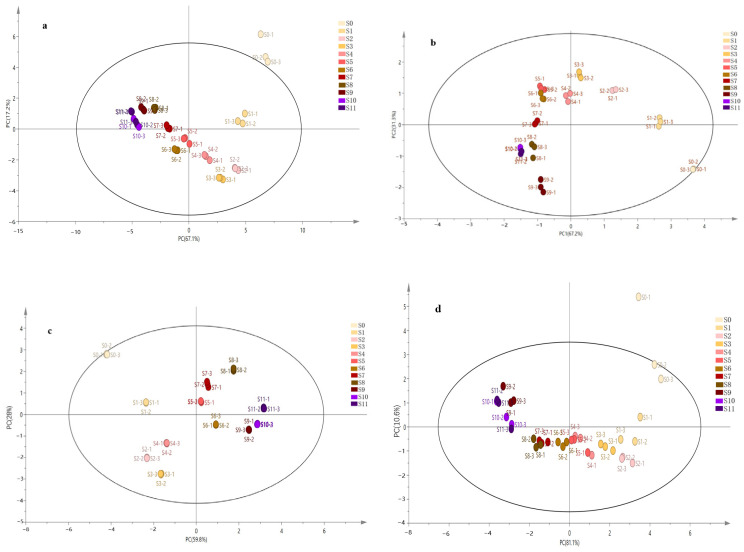
Analysis of PCH intelligent sensory technology after fitting. (**a**) PCA after fitting; (**b**) electronic eye; (**c**) electronic nose; (**d**) electronic tongue.

**Figure 5 foods-13-01586-f005:**
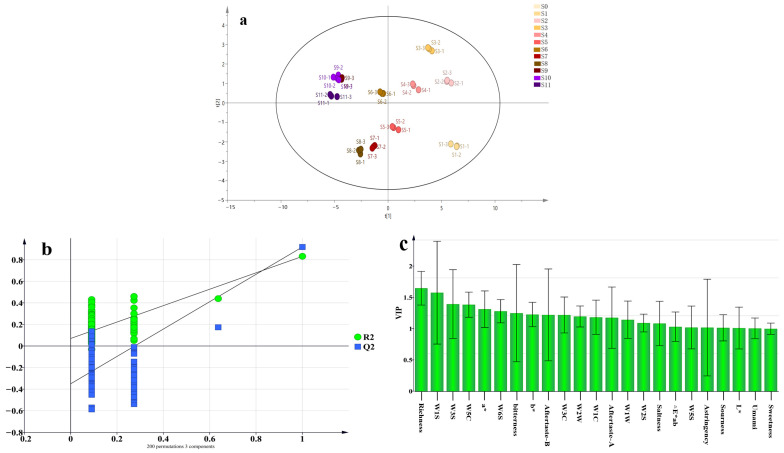
OPLS-DA of PCH intelligent sensory technology after fitting. (**a**) OPLS-DA scatter plot after removing outliers; (**b**) permutation plot of OPLS-DA; (**c**) VIP values of samples.

**Figure 6 foods-13-01586-f006:**
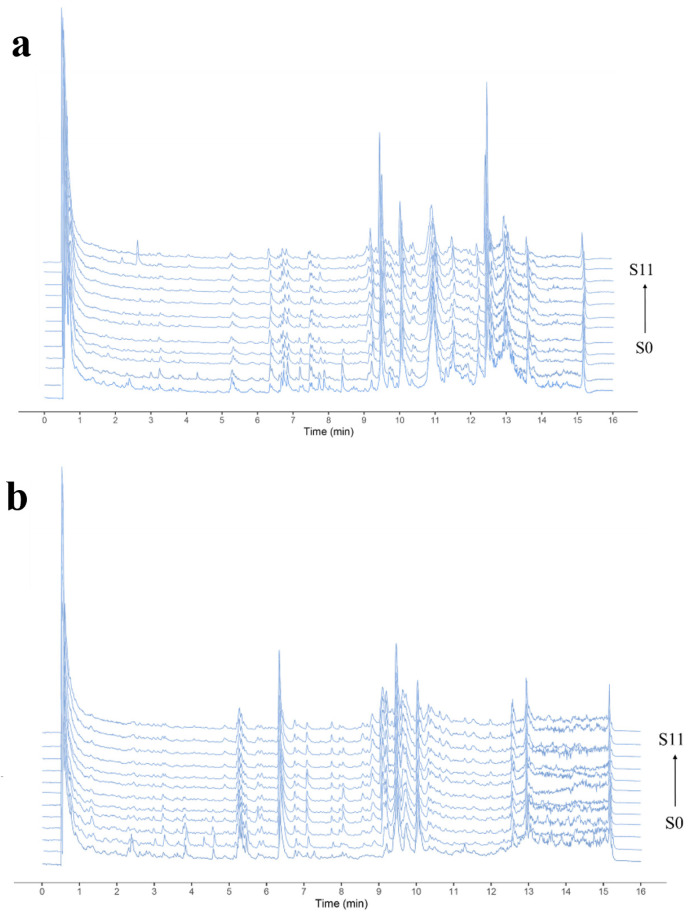
Total ion chromatogram of PCH extraction solution in both positive ion (**a**) and negative ion (**b**) modes.

**Figure 7 foods-13-01586-f007:**
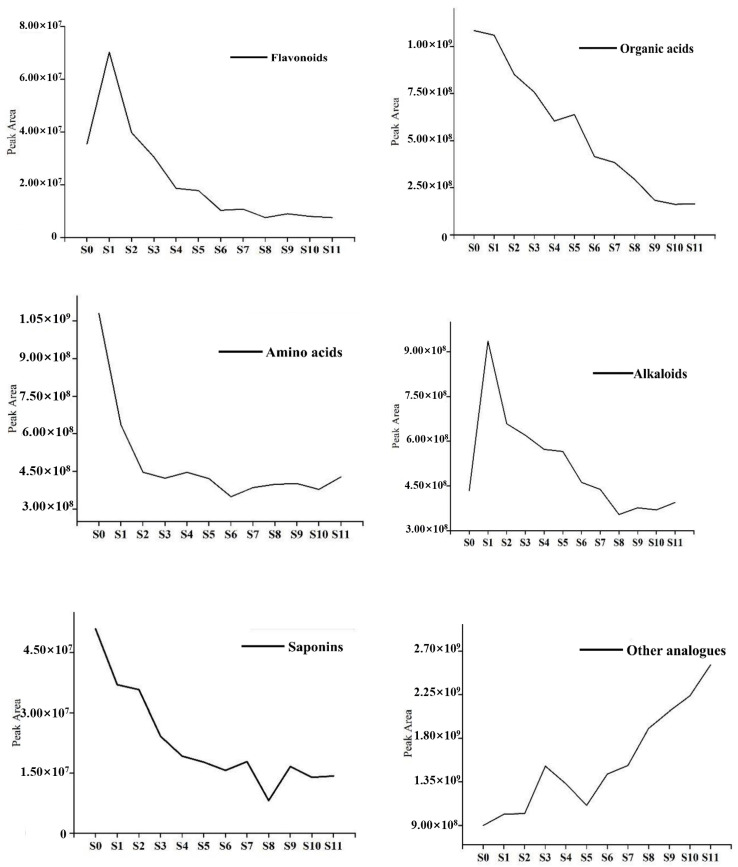
Changes in chemical components of PCH extraction solution in the process of industrial steaming.

**Figure 8 foods-13-01586-f008:**
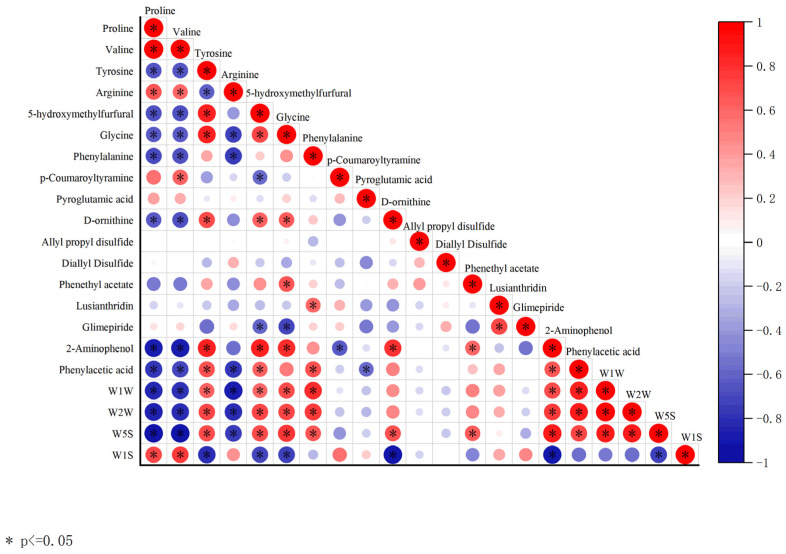
Correlation matrix heat map. Heat map of Spearman correlation between odour and HPLC analysis for each biomarker for the overall population analysed. * indicates significant differences between instrumental and odour parameters according to Pearson correlation coefficient (*p* ≤ 0.05).

**Figure 9 foods-13-01586-f009:**
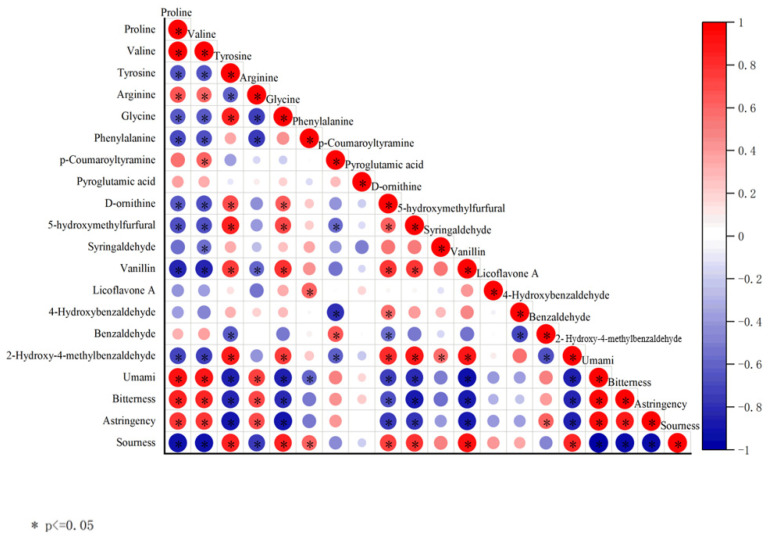
Correlation matrix heat map. Heat map of Spearman correlation between flavour and HPLC analysis between each biomarker for the overall population analysed. * indicates significant differences between instrumental and flavour parameters according to Pearson correlation coefficient (*p* ≤ 0.05).

**Table 1 foods-13-01586-t001:** The sensory evaluation criteria of PCH.

Item	Scoring Rules	Points
Colour (25 cents)	Uniformly dark and of uniform colour, with a glossy finish	18~25
	Blackish brown, largely homogeneous in colour, slightly discoloured	10~17
	Yellowish in colour, uneven and poorly coloured	1~9
Texture (20 cents)	Slightly harder and tougher, better texture	16~20
	Harder or softer, average texture	10~15
	Very hard or very soft, poor texture	1~9
Odour (25 cents)	Flavourful and sweet PCH	18~25
	Slightly sweet flavour of PCH	10~17
	No sweet smell of PCH	1~9
Taste (30 cents)	Sweet taste, not sticky to the teeth, good texture	20~30
	Sweeter flavour, more sticky to the teeth, average texture	11~20
	Unsweet or bitter taste, poor taste	1~10

**Table 2 foods-13-01586-t002:** Sensitive substance type of each sensor.

No.	Sensor Name	Sensitive Substance
1	W1C	Aromatic compounds
2	W5S	Nitrogen oxides
3	W3C	Ammonia, aromatic molecules
4	W6S	Hydrocarbons
5	W5C	Olefins, aromatic, polar molecules
6	W1S	Alkanes
7	W1W	Sulphur compounds
8	W2S	Alcohols, some aromatic compounds
9	W2W	Aromatic compounds, organic compounds of sulphur
10	W3S	Alkanes and aliphatics

**Table 3 foods-13-01586-t003:** Electronic tongue sensor information table.

Sensor	Corresponding Taste	Taste Information	
		Pre-Taste	Aftertaste
C00	Bitterness	Bitterness	Aftertaste-B
AE1	Astringency	Astringency	Aftertaste-A
CA0	Sourness	Sourness	-
CT0	Saltness	Saltness	-
AAE	Umami	Umami	Richness
GL1	Sweetness	Sweetness	-

Note: “-“ is a taste sensor that is not involved.

**Table 4 foods-13-01586-t004:** Sensory evaluation description of PCH industrialised steaming.

Sample	Colour	Texture	Odour	Taste	Score
S0	Yellowish White	Hard	No special odour	Bitter, astringent, tongue numbing	41
S1	Brown	Firm and hard	Slightly sweet	Slightly sweet	48
S2	Tan	Slightly hard and tough	Slightly sweet	Slightly sweet	63
S3	Tan	Slightly hard and tough	Sweet	Slightly sweet	65
S4	Black	Soft and sticky	Sweet	Sweet	69
S5	Ebony	Soft and sticky	Burnt flavour	Sweet	72
S6	Dark	Soft and sticky	Burnt flavour	Sweet	73
S7	Dark	Soft and sticky	Burnt flavour	Sweet	74
S8	Dark	Soft and sticky	Burnt flavour	Sweet	70
S9	Black	Soft and sticky	Burnt flavour	Slightly bitter, sour	67
S10	Black	Soft and sticky	Scorched flavour	Slightly bitter, sour	65
S11	Black	Soft and sticky	Scorched flavour	Slightly bitter, sour	62

**Table 5 foods-13-01586-t005:** Identification of chemical components in PCH extraction solution.

No.	RT (min)	Adduct Ions	ppm	Formula	Measured Value	Peak Intensity (*m*/*z*)	Chemical Compound	Source	References
1	0.4464	[M + H]^+^	2.084	C_6_H_10_S_2_	147.0286935	85.028; 147.028; 55.018; 57.033; 73.029	Diallyl Disulfide	f	[22]
2	0.5488	[M + H]^+^	4.659	C_5_H_9_NO_2_	116.070459	70.065; 56.049; 71.068; 44.049	Proline	d	[23]
3	0.5518	[M − H]^−^	0.608	C_6_H_8_O_7_	191.020116	111.01; 85.03; 87.009; 57.035	Citric acid	b	[24]
4	0.5518	[M − H]^−^	3.853	C_12_H_22_O_11_	341.108686	59.014; 89.025; 179.056; 101.024	Sucrose	f	[25]
5	0.5569	[M + H]^+^	1.227	C_5_H_11_NO_2_	118.086145	72.08; 119.049; 91.054; 59.073	Valine	d	[26]
6	0.5585	[M + H]^+^	0.302	C_9_H_11_NO_3_	182.081055	81.033; 136.075; 96.044; 165.054	Tyrosine	d	[23,27]
7	0.5586	[M + H]^+^	0.108	C_9_H_11_NO_2_	166.085982	167.088; 120.08; 148.076; 94.065	Xizanglongchine	c	[28]
8	0.6187	[M + H]^+^	0.180	C_6_H_14_N_4_O_2_	175.118968	70.065; 60.055; 116.071; 130.098	Arginine	d	[23,27]
9	0.6753	[M + H]^+^	0.509	C_6_H_6_O_3_	127.038935	109.029; 81.033; 53.038; 43.018	5-Hydroxymethylfurfural	f	[29]
10	0.8191	[M + H]^+^	1.679	C_8_H_8_O_4_	169.049284	170.079; 111.044; 97.028; 125.059	Vanillic acid	b	[30]
11	0.9395	[M + H]^+^	0.421	C_7_H_7_NO_2_	138.054942	120.044; 139.038; 110.06; 137.108	Glycine	d	[31]
12	0.9642	[M − H]^−^	1.3977	C_6_H_7_NO	108.045849	108.046; 109.03; 67.03; 65.014; 60.017	2-Aminophenol	f	[32]
13	0.9922	[M + H]^+^	3.032	C_8_H_12_N_2_	137.107416	138.055; 43.018; 120.045; 81.07	Chuanxiongzine	c	[28]
14	1.0482	[M − H]^−^	0.457	C_12_H_16_O_7_	271.083124	69.035; 71.015; 123.057; 272.058	Arbutin	b	[33]
15	1.1491	[M + H]^+^	0.253	C_9_H_10_O_4_	183.065046	184.095; 137.059; 43.018; 123.044	Eugenol	f	[34]
16	1.3355	[M − H]	1.225	C_7_H_6_O_2_	121.0298517	121.03; 122.024; 94.03; 120.023; 92.027	4-Hydroxybenzaldehyde	f	[35]
17	1.5806	[M − H]-	3.25	C_8_H_8_O_2_	135.045561	135.045; 44.998; 75.009; 93.035; 136.039	Phenylacetic acid	f	[36]
18	1.6133	[M + H]^+^	1.29	C_8_H_8_O_3_	153.0548033	153.055; 111.044; 125.06; 93.034; 65.039	Vanillin	f	[37]
19	2.0690	[M + H]^+^	0.153	C_20_H_22_O_4_	327.15505	168.065; 148.112; 281.15; 328.121	Dehydrodiisoeugenol	f	[38]
20	2.1524	[M − H]^−^	1.563	C_9_H_11_NO_2_	164.072256	165.056; 149.048; 121.031; 120.045	Phenylalanine	d	[23,27,39]
21	2.3004	[M + H]^+^	0.002	C_7_H_6_O_5_	171.029	172.075; 153.055; 127.03; 67.055	Gallic acid	b	[40]
22	2.3578	[M + H]^+^	2.850	C_12_H_14_O_5_	239.091681	83.049; 111.044; 95.049; 139.04	3,4,5-Trimethoxycinnamic acid	f	[41]
23	2.6248	[M + H]^+^	0.296	C_18_H_19_NO_5_	330.133902	312.12; 95.049; 193.051; 178.088	N-Feruloyloctopamine	c	[42]
24	2.6828	[M + H]^+^	0.402	C_16_H_14_O_6_	303.086878	151.038; 123.044; 81.034; 193.051	5,3′,5′-Trihydroxy-7-methoxyflavanone	a	[23,43]
25	2.6998	[M − H]^−^	0.855	C_7_H_6_O_3_	137.024883	93.035; 94.03; 138.02; 109.03	Salicylic acid	b	[44]
26	2.8531	[M + HCOO]^−^	0.304	C_56_H_92_O_29_	1273.56039	1273.482; 228.732; 1272.5; 1227.563	Digitonin	e	[45]
27	3.0016	[M + H]^+^	0.692	C_17_H_17_NO_3_	284.128196	148.048; 149.05; 147.045; 122.068	p-Coumaroyltyramine	c	[46]
28	3.1143	[M − H]^−^	0.674	C_24_H_34_N_4_O_5_S	489.2135268	59.014; 489.274; 71.015; 61.988; 101.024	Glimepiride	f	[47]
29	3.4300	[M + H]^+^	2.848	C_17_H_21_NO_3_	288.159179	270.147; 72.08; 81.034; 289.07	Piperine (chemistry)	c	[48]
30	3.4712	[M + H]^+^	2.854	C_15_H_10_O_6_	287.055181	107.05; 181.087; 288.056; 153.018	Kaempferol	a	[23]
31	3.6526	[M + H]^+^	1.000	C_8_H_8_O_2_	137.0597644	137.06; 81.07; 138.055; 43.018; 109.065	2-Hydroxy-4-methylbenzaldehyde	f	[49]
32	3.8455	[M + H]^+^	0.753	C_5_H_7_NO_3_	130.049902	86.096; 97.028; 84.043; 73.028	Pyroglutamic acid	d	[27,50]
33	3.9305	[M + H]^+^	2.665	C_20_H_18_O_4_	323.127139	291.104; 111.044; 81.034; 277.148	Licorice flavonoid A	a	[51]
34	3.9948	[M + H]^+^	5.659	C_57_H_96_O_28_	1229.60304	228.739; 414.313; 1230.374; 251.18	Sarsasapogenin	e	[52]
35	4.0237	[M + H]^+^	0.670	C_16_H_12_O_5_	285.075809	144.101; 81.069; 173.06; 183.114	Scutellarin	a	[53]
36	4.1517	[M − H]^−^	0.411	C_15_H_12_O_5_	271.062111	151.005; 119.05; 107.014; 65.004	Naringin	a	[20]
37	4.2184	[M + H]^+^	2.453	C_5_H_12_N_2_O_2_	133.101327	97.028; 134.06; 105.069; 69.033	D-Ornithine	d	[54]
38	4.3322	[M − H]^−^	2.478	C_8_H_6_O_4_	165.019591	121.03; 165.837; 93.035; 136.932	Piperonylic acid	b	[55]
39	4.4538	[M + H]^+^	1.904	C_10_H_10_O_3_	179.070341	133.065; 180.064; 161.059; 105.07	Coniferaldehyde	f	[56]
40	5.6975	[M − H]^−^	1.454	C_21_H_22_O_9_	417.120393	227.059; 165.056; 66.114; 105.02	Glycyrrhizin	a	[57]
41	6.0376	[M + H]^+^	1.1855	C_15_H_14_O_3_	260.1276919	260.126; 81.034; 106.05; 214.086; 59.37	Lusianthridin	f	[58]
42	6.0896	[M − H]^−^	1.260	C_19_H_18_O_6_	341.10357	279.199; 221.046; 59.014; 89.025	Methyl maltoflavanone A	a	[59]
43	6.2597	[M − H]^−^	0.918	C_20_H_27_NO_11_	456.146581	64.149; 57.115; 70.966; 72.457	Amygdalin	b	[60]
44	6.3256	[M + HCOO]^−^	5.294	C_48_H_78_O_20_	1019.5046	1019.473; 228.736; 116.93; 155.315	Madecassoside	e	[61]
45	6.4067	[M + H]	6.666	C_8_H_8_O_2_	137.0590863	137.059; 81.069; 138.054; 109.064; 95.085	Benzaldehyde	f	[62]
46	6.9704	[M + HCOO]^−^	1.004	C_42_H_72_O_14_	845.489849	845.518; 387.301; 228.736; 482.586	Ginsenoside Rf	e	[63]
47	7.5657	[M − H]^−^	1.799	C_15_H_10_O_5_	269.045516	225.055; 59.014; 270.049; 78.496	Apigenin	a	[64]
48	7.9753	[M + H]^+^	3.291	C_18_H_30_O_2_	279.231081	69.07; 83.085; 95.085; 71.085; 81.069	α-Linolenic acid	f	[65]
49	8.4767	[M + H]^+^	2.986	C_18_H_32_O_2_	281.24716	263.144; 221.134; 235.147; 135.079	Linoleic acid	f	[65]
50	8.6366	[M + H]^+^	1.706	C_27_H_42_O_3_	415.320291	271.207; 149.096; 253.195; 72.044	Diosgenin	e	[66]
51	10.9008	[M + H]^+^	2.867	C_10_H_12_O_2_	165.0905267	121.065; 166.086; 93.07; 165.09; 91.054	Phenethyl acetate	f	[67]
52	10.8588	[M − H]^−^	0.113	C_18_H_32_O_2_	279.233032	280.239; 59.013; 261.224; 134.895	Linolenic acid	f	[65]
53	11.1044	[M − H]^−^	6.410	C_48_H_78_O_17_	925.504068	152.996; 925.481; 279.234; 255.235	Saikosaponin BK1	e	[68]
54	11.5801	[M − H]^−^	2.366	C_16_H_32_O_2_	255.233604	256.237; 219.846; 119.05; 69.793	Palmitic acid	f	[69]
55	11.9280	[M + H]^+^	1.557	C_6_H_6_O_3_	127.038802	109.029; 81.034; 43.018; 128.143	Maltol	a	[70]
56	11.9681	[M + H]^+^	2.975	C_6_H_12_S_2_	178.034224	149.024; 108.045; 167.054; 126.054; 169.034	Allyl propyl disulfide	f	[71]
57	12.5736	[M + H]^+^	1.413	C_9_H_11_NO_2_	166.085765	136.075; 120.08; 80.049; 167.088	Polygonatine A	c	[72]
58	12.5887	[M − H]^−^	3.468	C_12_H_18_O_3_	209.118275	126.02; 210.077; 183.068; 124.041	Jasmonic acid	f	[73]
59	12.8387	[M − H]^−^	2.895	C_36_H_62_O_9_	637.431155	71.015; 89.025; 279.233; 101.024	Ginsenoside F1	e	[74]
60	12.9707	[M + H]^+^	0.723	C_10_H_13_N_5_O_4_	268.103806	136.061; 67.054; 81.069; 76.28	Adenosine triphosphate	f	[75]
61	12.9805	[M + H]^+^	4.429	C_5_H_5_N_5_	136.061397	137.083; 109.077; 81.069; 95.06	Adenine	f	[76]
62	13.4751	[M + H]^+^	1.401	C_16_H_22_O_4_	279.158609	149.024; 81.069; 67.054; 95.086	Dibutyl phthalate	b	[77]
63	13.7474	[M + H]^+^	0.684	C_9_H_10_O_5_	199.180136	74.097; 200.129; 100.076; 53.456	Butyric acid	b	[78]
64	15.6257	[M + H]^+^	0.338	C_5_H_11_NO_2_	118.08604	72.08; 59.073; 119.09; 58.065	Betaine	c	[79]

Note: a: flavanols; b: organic acids; c: alkaloids; d: amino acids; e: saponins; f: others.

## Data Availability

The original contributions presented in the study are included in the article/Appendix A, further inquiries can be directed to the corresponding authors.

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
