# Peer review of "Quality Evaluation of Polygonatum cyrtonema Hua Based on UPLC-Q-Exactive Orbitrap MS and Electronic Sensory Techniques with Different Numbers of Steaming Cycles"

_foods, 2024, doi:10.3390/foods13101586_

Round 1
Reviewer 1 Report
Comments and Suggestions for Authors
Comments on the Quality of English LanguageAuthor Response
Please check the attacchment.

Reviewer 2 Report
Comments and Suggestions for Authors
The authors present a combination of sensory evaluation by an assessment panel with electronic sensory devices and LC-MS based metabolomics to evalutate the processing procedure of a medicinal plant. However, despite the exhaustive use of chemometric procedures and correlation analyses, the overall benefit of the presented work remains unclear. My major concern is that there is no correlation of the electronic sensor data with the sensory evaluation of the assessment panel. This would be needed to highlight the the usability of the electronic sensors in lieu of the human panel. Moreover, even though LC-MS based analyses oftentimes provide valuable insights in the context of food quality evaluation, in this case neither the electronic sensory data nor the the metabolomics data have been convincingly used to provide additional information on top of the sensory panel assessment.
In addition, there are several more flaws and errors throughout the manuscript:
- There is insufficient information with regard to the employed steaming equipment and used temperature program
- No/insufficient information is given on the used 'electronic eye' and 'electronic nose' instrumentation (e.g. vendor, instrument type)
- Table 4: The scores of the sensory evaluation panel should be given in the table as well
- Figures 2 and 4: No error bars are shown despite the experiments having been performed in triplicate
Figures 3, 6 and 7: The readability of the graphs would improve if a colour gradient was used rather than a random colour for every sample
- Insufficient information is given on the hierarchical cluster analysis. WHich distance metric and linkage type was used? There should be an additional statistics chapter in the method section, also stating which software was used for the chemometric analyses
- Figure 5: Axis label has not been translated to English
- Figure 6: sub-figure e) is described in the caption but is missing
- Lines 309 - 313: The conclusions in the text don't line up with the data depicted in Figure 7c. Some of the mentioned descriptors clearly have a VIP value < 1 in the figure.
- Figure 7b: Axis-labels are missing or unreadable-
- Figure 8: y-axis and label missing
- Table 5: Compounds 1 and 2 have an indicated retention time clearly shorter than the dead volume of the column (injection peak in figure 8 is around 0.5 min). How do you explain this?
- Section 3.2.2. is too exhaustive and should be shortened. Even though the authors cleary put some effort in identification of the compounds, use of dedicated software to match the spectra with database entries plus a stringent score cutoff would have probably been enough in the context of this work
Comments on the Quality of English LanguageUnfortunately, quality of the English language in this manuscript is relatively poor and clearly hampers the readability. Appart from grammatical and orthographic errors, there are numerous fragmented sentences (e.g. already the first sentence in the abstract or lines 187-189). In the current state, the manuscript is unfit for publication.
Round 2
Reviewer 2 Report
Comments and Suggestions for Authors
The authors thoroughly worked through the criticisms and thus considerably improved both readability and scientific soundness of the manuscript. In the opinion of this reviewer the manuscript is fit for publication now.